# Challenges for Immunotherapy in Multiple Myeloma: Bone Marrow Microenvironment-Mediated Immune Suppression and Immune Resistance

**DOI:** 10.3390/cancers12040988

**Published:** 2020-04-17

**Authors:** Lisa C. Holthof, Tuna Mutis

**Affiliations:** Department of Hematology, Amsterdam UMC, VU University Medical Center, Cancer Center Amsterdam, 1081HV Amsterdam, The Netherlands; l.holthof@amsterdamumc.nl

**Keywords:** multiple myeloma, immunotherapy, microenvironment, immune resistance, immunosuppression, apoptosis resistance, drug resistance, CAR T-cells, monoclonal antibodies

## Abstract

The power of immunotherapy in the battle of Multiple Myeloma (MM) started with allogeneic stem cell transplantation, and was rediscovered with immunomodulatory drugs and extended with the outstanding results achieved with targeted antibodies. Today, next to powerful antibodies Elotuzumab and Daratumumab, several T-cell-based immunotherapeutic approaches, such as bispecific antibodies and chimeric antigen receptor-transduced T-cells (CAR T-cells) are making their successful entry in the immunotherapy arena with highly promising results in clinical trials. Nonetheless, similar to what is observed in chemotherapy, MM appears capable to escape from immunotherapy, especially through tight interactions with the cells of the bone marrow microenvironment (BM-ME). This review will outline our current understanding on how BM-ME protects MM-cells from immunotherapy through immunosuppression and through induction of intrinsic resistance against cytotoxic effector mechanisms of T- and NK-cells.

## 1. Introduction

Multiple Myeloma (MM), the malignant disease of monoclonal, antibody-producing plasma cells in the bone marrow (BM), is the second most common hematological malignancy, accounting for 20% of deaths from hematological malignancies. For decades, the standard therapy of MM was based on high-dose chemotherapy with alkylating agents, mainly melphalan, combined with autologous transplantation. Currently, new chemotherapeutic agents are available for the treatment of MM including second- and third-generation proteasome inhibitors carfilzomib and ixazomib, and histone deacetylase inhibitors panabinostat and vorinostat. However, even “low-risk” patients do not remain in long-lasting remissions after traditional or novel MM treatments [1,2,3]. Due to their high genetic instability and the support from the BM microenvironment (BM-ME), MM-cells rapidly develop resistance to virtually all chemotherapies developed so far [1,2,3,4]. To date, the only MM therapy with curative potential in a fraction of patients is allogeneic stem cell transplantation. The allo transplantation can eradicate MM-cells due to the well-known graft versus Myeloma effect, which is predominantly mediated by donor T-cells present in the graft. However, this unspecific form of allogeneic immunotherapy is no longer the first choice of treatment, especially for “low and standard” risk patients, due to high rates of transplant-related mortality and morbidity. Nonetheless, the “allogeneic transplantation practice” clearly illustrated immunotherapy could be a curative option for MM patients, if it can be made selective for MM-cells. In fact, starting from the late nineties, immunotherapy strategies have been successfully implemented in MM treatment. The sequential introduction of immunomodulatory drugs (IMiDs) including thalidomide, lenalidomide and pomalidomide in MM treatment had a significant positive impact on the life expectancy of patients who relapsed from standard chemotherapies. While patients appeared to develop resistance against direct anti-MM effects of IMiDs, several analyses revealed that their T- and NK-cell activating properties remained largely intact, making IMiDs ideal partners for combination immunotherapies [5,6,7]. IMiDs were rapidly followed by highly successful antibodies such as the SlamF7-specific Elotuzumab and the CD38-specific Daratumumab. These antibodies achieve unprecedented response rates in heavily pretreated patients, especially in combination with IMiDs and proteasome inhibitors [8]. 

Currently, much effort is being devoted to additionally exploit the full cytotoxic power of T-cells against MM by the development of T-cell-engaging bispecific antibodies [9], MM-specific-alpha/beta or gamma-delta T-cells [10], chimeric antigen receptor (CAR)-transduced T-cells [11,12] and vaccines to prime and activate MM-specific autologous T-cells immunotherapy [13]. Nonetheless, similar to the observations in several other cancers, the responses of MM patients to immunotherapy are not long lasting, indicating that MM is also able to escape from these potentially very powerful immunotherapy strategies. 

The ultimate success of immunotherapy in MM and other cancers will largely depend on unraveling and effective modulation of important immune escape mechanisms. Extensive research in the past decade already revealed the highly immunosuppressive nature of the MM BM-ME. Furthermore, we and other investigators have discovered that the anti-apoptotic mechanisms, which are significantly upregulated by tight cellular interactions in the BM-ME, can induce an intrinsic resistance in MM-cells towards cytotoxic mechanisms of immune cells. This review will mainly focus on the recent findings on the BM-ME-induced immune resistance, after an overview of the immunosuppressive mechanisms in the MM BM-ME. 

## 2. Immunosuppression and Immune Exhaustion in Bone Marrow Microenvironment 

The gradual transformation of the asymptomatic monoclonal gammopathy of undetermined significance (MGUS) into to symptomatic MM is associated with increased genetic mutations but also with significant changes in the cellular composition of the BM-ME and the subsequent loss of functional immune surveillance [14]. These cellular changes involve the development and/or recruitment of various immunosuppressive cells, including myeloid derived suppressor cells (MDSCs), regulatory T-cells (Tregs), regulatory B-cells (Bregs) and tumor-associated macrophages (TAMs) in the BM-ME (Figure 1). 

Among these cell subsets, MDSCs are a heterogeneous population of immature myeloid cells, phenotyped as CD33^+^CD11b^+^ but HLA-DR^dim/neg^ cells within CD14^+^ monocytic or CD15^+^ granulocytic lineages. MDSC frequencies gradually increased in the BM-ME during MM development, reaching highest levels in relapsed and/or refractory MM patients [15]. Especially driven by the activation of the STAT3 pathway that is stimulated through cytokines like interleukin 6 (IL-6) and vascular endothelial growth factor (VEGF), MDSCs possess the ability to suppress immune responses via a number of mechanisms involving the secretion of nitric oxide (NO), arginase, reactive oxygen species (ROS), prostaglandin E2 (PGE2), or Indoleamine 2,3-Dioxygenase (IDO). These immune suppressive cytokines can inhibit the proliferation and expansion of Th1-cells, cytotoxic T-lymphocytes (CTLs) and NK-cells, and facilitate the differentiation and the recruitment of TH17-cells, Tregs as well as TAMs in the microenvironment [15,16]. 

Tregs, characterized as CD4^+^CD25^+^Foxp3^+^CD127^dim/−^ T-cells, are undoubtedly the most extensively studied immunosuppressive cell subset in cancer immunology and in the context of MM. Tregs are well known to inhibit Th1, Th17, CTL, macrophage, and DC function by cellular interactions and via secretion of suppressive cytokines, such as transforming growth factor beta (TGF-β) and IL-10 [17]. The frequencies of Tregs gradually increase in the BM-ME from progression of MGUS to overt MM disease [18,19,20,21]. Conversely, it was shown that the frequencies of Tregs decrease in MM patients after successful treatment with lenalidomide plus dexamethasone [18,19,20]. Moreover, we found inverse correlations between frequencies of recipient Tregs and their response to donor lymphocyte infusions [22]. For Tregs, the STAT3 pathway plays an important role in cellular development, proliferation and function, which is achieved through upregulating the transcriptional expression of the hallmark of Tregs, FOXP3 [23]. Interestingly, a common feature of all immunosuppressive cells, including Bregs and TAMs, is that they all express high levels of CD38, which can be targeted by CD38-directed antibodies including Daratumumab and Isotixumab [24,25]. We have recently shown that Daratumumab therapy rapidly depletes CD38^+^ Tregs, MDSCs and Bregs in the peripheral blood and in the BMand is associated with clonal expansion of CD4^+^ and CD8^+^ T-cells in MM patients [25]. Thus, it seems possible to use CD38-directed antibodies to target not only MM-cells but also immunosuppressive cells, to restore the preexisting anti-MM T-cell responses in the BM-ME even after immune checkpoint blockade [26].

Next to these suppressive immune cells, mesenchymal stromal cells (MSCs) and osteoclasts also significantly contribute to an immunosuppressive environment (Figure 1). It has been shown that the crosstalk between MM-cells and MSCs mediated by toll-like receptor 4 (TRL4) signaling transforms MSCs into a “malignant” phenotype, which promotes tumor growth and immune escape [27]. The MM-conditioned MSCs have been shown to suppress T-cell activation and proliferation, impair DC maturation and induce Tregs via the secretion of several factors, including IL-6, TGF-β, IL-10, PGE2, and upregulated expression of several surface molecules such as VCAM-1, intracellular adhesion molecule-1 (ICAM-1) and CD40 [27,28,29,30,31,32]. Additionally, MSCs can exert their immunomodulatory activities by the secretion of extracellular vesicles [33,34]. Osteoclasts and other myeloid cells contribute to an immunosuppressive environment especially by the production of A proliferation inducing ligand (APRIL), which is the ligand for BCMA that is expressed on MM-cells and for TACI that is expressed on MM-cells and Tregs. APRIL not only facilitates MM-cell growth and survival but also stimulates the upregulation of TGF-β and IL-10 in the BM-ME [35] and promotes the survival of Tregs via TACI signaling [36].

Finally, the immune suppressive BM-ME involves strong upregulation of important immune checkpoint molecules during the transition of MGUS to malignant MM (Figure 1). Expression of immune checkpoint molecule-programmed cell death 1 (PD-1) on effector T- and NK-cells, and its ligands PD-L1/2 on MM-cells [37,38,39], is well-known to be induced and enhanced through an immune-mediated IFN-γ response [40]. The PD-L1/2 expression on MM-cells can also be promoted through the stimulation of TLR ligands [40], interactions with MSCs [41] or signaling via APRIL [35]. While not much is known about the involvement of immune checkpoints Lag3 and TIM3 in MM progression, the T-cell immunoglobulin and ITIM domain (TIGIT), which is expressed on both T- and NK-cells [42], is perhaps another important immune checkpoint in MM. It has recently been shown that progression of MM was associated with high levels of TIGIT expression on CD8^+^ T-cells, which displayed impaired proliferative and cytokine responses upon non-specific and MM-antigen stimulation [43]. Current clinical trials are evaluating the effects of antibody-mediated blockade of immune checkpoints alone or in combination with IMiD therapies. Although immune checkpoint blockade only is not effective in heavily pretreated relapsed and/or refractory MM patients [44,45], combination with lenalidomide plus dexamethasone was more successful in a phase I trial [46]. Nonetheless, in randomized phase three trials, the benefit–risk profile of PD-1 blockade plus lenalidomide and dexamethasone was unfavorable, both in newly diagnosed and relapsed and/or refractory MM patients [47,48], raising questions about future targeting of PD-1 and PD-L1 in MM [49]. Possibly, additional modulation of the immune suppressive tumor microenvironment is required. For example, a recent study introduced dual targeting of PD-1 and receptor activator of nuclear factor kappa-B ligand (RANKL) to simultaneously block immune checkpoint-mediated immune suppression and reduce osteoclast formation that, thereby, may increase immunotherapeutic anti-MM activity [50].

## 3. Immune Escape Mechanisms beyond Immune Suppression and Immune Exhaustion

Over the past decade, considerable studies focused on immune suppression and immune exhaustion within the tumor ME. Consequently, many researchers consider these two modes of immune modulation as the major, if not the only, mechanisms of tumor immune escape. Obviously, there are also many other, “tumor intrinsic” mechanisms of immune escape such as downregulation of MHC and/or costimulatory molecules, blockade of antigen processing machinery by viral tumors, loss or mutation of the target antigen and upregulation of decoy receptors or complement inhibitory receptors. All of these alterations in tumor cells will result in diminished effector T/NK-cell or complement function without active immune suppression or exhaustion. While the role of the tumor ME in these mechanisms of immune escape is not well studied, others and we have recently discovered and pointed out the potential importance of another important mechanism of immune escape in MM, which is strictly induced by BM-ME (Figure 1 and Figure 2). This type of “BM-ME-mediated immune resistance” is also “tumor intrinsic” and is established by the intensive cross-talk of MM-cells with the cells of the BM-ME via soluble factors such as IL-6, APRIL and growth factors, but most importantly via the integrin-mediated cell adhesion and Notch signaling (Figure 1 and Figure 2). At the molecular level, the BM-ME-mediated immune resistance is highly associated with the inhibition of apoptosis [51] (Figure 2). Therefore, the next section will outline the main pathways of apoptosis and how they are involved in the targeted cell lysis that is induced by immune killer cells including CD4^+^ and CD8^+^ CTLs and NK-cells. 

## 4. Apoptosis, the Main Mechanism of Immune Cell-Mediated Tumor Cell Lysis

Apoptosis, the programmed cell death, is a normal cellular event that occurs in countless amounts of cells every day. It is the result of an irreversible cascade of molecular events, finally leading to DNA fragmentation and nuclear blebbing (Figure 2). The major mechanisms of apoptosis induction are mediated by caspases. There are two major pathways to initiate caspase-dependent apoptosis: i) through the stimulation of death receptors on the cell surface (extrinsic pathway) and ii) through internal stress signals resulting in mitochondrial destabilization (intrinsic pathway). The extrinsic apoptotic signaling starts with the ligand-induced clustering of one of the death receptors TNFR1, FAS, DR4 [TRAILR1], DR5 [TRAILR2], or DR6. Once clustered, the death receptors undergo allosteric conformational changes and bind the adaptor protein FADD, which subsequently activates pro caspase-8. With the increased enzymatic activity, caspase-8 cleaves and activates pro caspases-3 and -7, which are known as executioner caspases that can initiate DNA fragmentation in the nucleus. The extrinsic pathway also intersects with the intrinsic pathway since caspase-8 is also capable of activating the BID protein, an important initiator of the intrinsic pathway [52]. The intrinsic pathway involves the disruption of the mitochondrial membrane. This is established by BID- or BIM-mediated activation of pro-apoptotic proteins BAK and BAX. The activated BAK and BAX are recruited to the mitochondrial membrane where they form complexes to destabilize the membrane. This results in the release of mitochondrial cytochrome C into the cytosol where it binds together with dATP to Adapter Protein Apoptotic Protease-activating Factor-1 (Apaf-1) to form the so called “apoptosome”, which subsequently recruits and activates pro caspase-9. Similar to caspase-8, activated caspase-9 is able to activate the executioner caspases-3 and -7. 

In our current understanding, induction of apoptosis is the main mechanism of CTLs and NK-cells to kill tumor-cells with some reported exceptions [53]. CTLs and NK-cells are known to utilize both the extrinsic and intrinsic pathways to induce apoptosis. Upon tumor-cell encounter, they can activate death-receptor mediated extrinsic pathway via FAS-ligand and/or TRAIL molecules expressed on their surface upon activation. Additional to CTLs and NK-cells, therapeutic death receptor antibodies can directly mediate death receptor signaling [54,55]. 

The other major cytotoxic machinery of CTLs and NK-cells is the degranulation of perforin and granzyme-containing cytotoxic granules within the immune synapse. Perforin generates holes in the target cell membrane and in the membranes of endosomes to aid the entry of granzymes in the cytoplasm. Once entered in the cytoplasm, granzymes initiate apoptosis by their serine protease activity. Among the five known human granzymes to date (granzymes -A, -B, -H, -K, and -M), granzyme-A and -B are most abundant in the cytotoxic granules of CD4^+^ and CD8^+^ CTLs and NK-cells. Additionally, NK-cells’ granules can also contain granzyme-M [56]. While granzyme-B can directly activate effector caspases-3 and -7, it preferentially mediates target cell killing by activating the intrinsic pathway of apoptosis through the cleavage of BIM and/or BID [57,58,59]. 

Besides these well-known, caspase-dependent pathways of apoptosis, there are also caspase-independent apoptosis pathways involving serine proteases, cathepsins and calpains [60]. One of the caspase-independent mechanisms can be activated by granzyme-A, which begins in the mitochondrion but does not involve BAK and BAX or cytochrome C release. Instead, granzyme-A can cleave the mitochondrial Complex I Protein Ndufs3, which disrupts mitochondrial metabolism and generates reactive oxygen species (ROS) [61]. ROS, in turn, initiates the translocation of endoplasmic reticulum-associated SET/DNase NM23-H1 complexes into the nucleus, where SET is degraded, allowing NM23-H1 to nick chromosomal DNA to induce single stranded DNA damage [62]. 

## 5. Inhibition of Apoptosis by the BM-ME

Over the past decades, intensive research addressing the role to the BM-ME in hematological malignancies clearly demonstrated that the accessory cells of the BM can support the survival and proliferation of leukemia and MM-cells not only by promoting growth but also by inhibiting apoptosis [63]. The anti-apoptotic effects of the BM-ME are primarily established by upregulation of anti-apoptotic regulatory proteins via the stimulation of RAS/MEK/ERK, JAK/STAT3, PI3K/Akt as well as the NF-kB signaling pathways [64,65,66]. For instance, multiple members of the BCL2 family of proteins BCL-2, BCL-X_L_ and MCL-1, which are known to negatively regulate BAK- and BAX-mediated mitochondrial membrane destabilization [67], are significantly upregulated by activation of the above mentioned signaling pathways via several soluble factors produced by BM MSCs such as IL-6, IGF-1 or VEGF [68]. Additionally, the abundantly present Insulin-like growth factor 1 (IGF-1) can downregulate the pro-apoptotic molecule BIM via activation of the AKT pathway [69]. Similar effects are also established by integrin-mediated adhesion of MSCs to MM-cells and other hematopoietic tumor-cells [70,71]. 

The cross-talk between tumor-cells with the microenvironment via soluble factors or cell–cell contacts are also known to significantly upregulate the Inhibitors of Apoptosis (IAP) family of anti-apoptotic proteins. The best-characterized members of this family are XIAP and Survivin (BIRC5), which are frequently overexpressed in human tumors. Both IAP proteins inhibit the executioner caspases-3 and -7, and XIAP additionally inhibits caspase-9. In the MM BM-ME, MSCs can upregulate both XIAP and Survivin in MM and other tumors [72,73,74,75,76] through the stimulation of NF-κB signaling [66,77], downregulation of the Survivin-targeting microRNA miRNA-101-3p [74] or through NOTCH signaling [78,79,80,81]. 

The BM-ME also influences the extrinsic apoptotic pathway as stroma-MM-cell interactions can significantly affect the expression of death receptors. Soluble factors secreted from BM MSCs are known to upregulate the FLICE-like inhibitory protein (c-FLIP) [82], which is an effective inhibitor of the caspase-8 activation by binding and reducing the availability of FADD. In addition, integrin-mediated cell adhesion increases cytosolic solubility of c-FLIP and facilitates its binding to FADD [83]. Finally, MSCs also significantly inhibits the caspase-independent apoptosis, that can be induced by granzyme-A, by reducing ROS levels and mitochondrial membrane potential via an upregulation of antioxidant production [84].

## 6. BM-ME-mediated Immune Resistance: The Proof of Concept

Concerning the aforementioned complex anti-apoptotic mechanisms, the BM-ME is known to induce perhaps the best documented form of epigenetic, environment-mediated resistance to several anti-cancer drugs [85]. Since the induction of apoptosis is the main mechanism of tumor-cell elimination by immunotherapeutic T- and NK-cells, it is plausible that the BM-ME also protects MM-cells from the cytotoxic machinery of CTLs and NK-cells. Over the past years, others and we have addressed this interesting possibility in in vitro [86,87,88] as well as in a unique in vivo xenograft model where human MM-cell lines were grown on human MSC-coated scaffolds [89]. In these model systems, we demonstrated that the cytotoxic activity of HLA-restricted, Myeloma-reactive CD4^+^ and CD8^+^ CTLs [86] and antibody-dependent cellular cytotoxicity (ADCC) mediated by Daratumumab [87] against MM-cells are significantly diminished in the presence of MSCs or human umbilical vein endothelial cells (HUVECs) without any signs of immune suppression or target antigen downregulation [86,87]. We designated this type of immune resistance as “cell adhesion-mediated immune resistance” (CAM-IR) because the protection from CTLs by MSCs was predominantly mediated by direct MSC–MM-cell adhesion, while soluble factors played only a minor role [86]. Further analyses revealed that MM-cells upregulated anti-apoptotic proteins Survivin and MCL-1 upon co-culture with stromal cells. Furthermore, the sh-RNA-mediated downregulation of Survivin or inhibition of Survivin and MCL-1 expression with a small inhibitory molecule, YM155, largely abrogated the protective effects of MSCs and synergistically upregulated the lysis of MM-cells by CTLs as well by Daratumumab in ADCC assays [86,87]. Hence, these experiments revealed, similar to BM-ME-mediated drug resistance, the proof of concept that MSCs are able to induce an intrinsic immune resistance in MM-cells through upregulation of anti-apoptotic proteins. 

In these studies, we demonstrated that, like cell adhesion-mediated drug resistance (CAM-DR), CAM-IR can be inhibited by blocking integrin binding on intact cells, but, unlike CAM-DR, CAM-IR cannot be induced by sole binding of MM-cells to fibronectin, vitronectin, or laminin on its own [86]. Immune resistance is therefore most likely triggered by a collective action of integrins and other receptor-ligand systems. Possibly, NOTCH signaling plays a role in CAM-IR as it does in CAM-DR [90]. Addressing this intriguing possibility, we indeed observed that inhibition of NOTCH signaling by gamma secretase inhibitors (GSI) can also abrogate CAM-IR and increase the cytotoxic activity in a synergistic manner (manuscript in preparation). Beyond these mechanisms, there might be other ways how BM-ME shields MM-cells from immune attack. For instance, IL-6 can promote an NF-κB-dependent increase in c-FLIP expression in MM-cells, which was shown to protect against recombinant TRAIL [91]. Furthermore, NF-κB signaling as well as hypoxia can induce expression of the serine protease inhibitor-9 (PI-9) in MM-cells [92,93], which inactivates granzyme-B and, thereby, can induce resistance against cytotoxic immune cell-mediated lysis [94,95].

## 7. BM-ME-Mediated Immune Resistance against CAR T-cells, Bispecific Antibodies and Death Cell Receptor Antibodies

Since the BM-ME-mediated immune resistance may have a great impact on novel immunotherapies, we recently extended these studies to investigate whether BM MSCs can also protect MM-cells from novel immunotherapeutic approaches such as CAR T-cells, bispecific antibodies and death receptor-mediated antibodies. In these studies, we observed that MSCs could protect MM-cells from highly lytic BCMA-CAR T-cells only at very low effector to target cell ratios, while they readily inhibited the MM-cell lysis by moderately lytic CD38-specific and CD138-specific CAR T-cells, which were generated using intermediate to low affinity antibodies. Overall, we demonstrated a strong inverse correlation between the avidity of CAR T-cells and the extent of MSC-mediated protection [96]. For T-cell engaging bispecific antibodies, we found similar results. While MSCs did not influence the lysis of MM-cells by highly lytic BCMA/CD3 bispecific antibodies [97], the somewhat lower cytotoxic activities of GPRC5/CD3 bispecific antibody was readily inhibited by MSCs [98]. Finally, when we tested a number of clinically applied DR5-antibodies, which exclusively activates the extrinsic apoptotic pathway with no involvement of effector killer cells, we observed that MSCs could readily inhibit MM-cell lysis induced by these antibodies [96]. These recent results strongly suggested that BM-ME-mediated immune resistance could be an important factor for the clinical outcome of novel immune therapies including CAR T-cells, T- or NK-cell engaging (bispecific) antibodies or even death cell-receptor antibodies which do not involve effector killer cells. 

## 8. Conclusions and Future Directions

Our recent studies illustrate that, besides immunosuppression and immune exhaustion, apoptosis resistance induced by cellular interactions is another important mechanism of immune evasion in the BM-ME. It is highly likely, and may even be imperative, that these three major mechanisms of BM-ME-mediated immune evasion should all be targeted in order to achieve successful and long-lasting outcomes from T- and NK-cell-based therapies. The modulation of BM-ME-mediated immune resistance is also highly important for antibody therapies, including antibody-drug conjugates, which are designed to directly induce apoptosis in MM-cells. In this context, a plausible option is to maximize the killing capacity of bispecific antibodies, death receptor antibodies and CAR T-cells by increasing their affinity to MM-cells. Another option could be to improve the overall avidity of T-cells toward MM-cells by endowing them with dual or multiple CARs, whereby more than one single MM-associated antigen can be targeted [99]. Improving the interaction between T-cells and MM-cells this way can also result in more efficient lysis levels, which is required to overcome the BM-ME-mediated resistance. In addition, CAR T-cells may benefit from the incorporation of the CD28 costimulatory domain in the CAR constructs, since this co-stimulatory domain improves the cytotoxic activity also of low affinity CAR T-cells [100]. However, when the target antigen is not entirely MM-cell specific, such is the case with CD38 or SlamF7, increasing the affinity of (bispecific) antibodies or CAR T-cells may increase the risk of off-tumor toxicity. Therefore, it may be preferred to modulate the BM-ME-mediated immune resistance by including pro-apoptotic agents such as inhibitors of MCL-1, BCL-2 or Survivin. As mentioned, we successfully modulated the MSC-mediated immune resistance using YM155, a small molecule inhibitor of Survivin and MCL-1. In more recent studies we used another novel small molecule, FL118, which inhibits multiple anti-apoptotic proteins including Survivin, MCL-1, and XIAP [101]. This small molecule, which is effective against multiple drug-resistant solid tumors, also demonstrates potent single agent anti-MM activity and was able to abrogate MSC-mediated drug resistance in our recent preclinical studies [102,103,104,105,106]. In our preliminary studies, FL118 is also able to largely abrogate the MSC-mediated immune resistance against HLA-restricted T-cells, CAR T-cells, Daratumumab and DR5 antibodies, without showing toxicity for immune cells [100] (manuscript in preparation). These results illustrate once again that there are converging mechanisms of BM-ME-induced drug and immune resistance and provide opportunities to effectively modulate immune resistance with agents that are already known to modulate drug resistance, such as FL118. Nonetheless, the converging mechanisms between BM-ME-induced drug resistance and immune resistance also suggest that heavily pretreated multidrug-resistant patients are not the ideal candidates for immunotherapy. Thus, towards cure of MM, the ideal design of a future clinical trial would not only be based on combination therapies that can tackle immune suppression, immune exhaustion as well as BM-ME-mediated resistance to apoptosis, but would also be applied in an earlier stage, long before the BM-ME can induce resistance in MM-cells against conventional and less effective drugs.

## Figures and Tables

**Figure 1 cancers-12-00988-f001:**
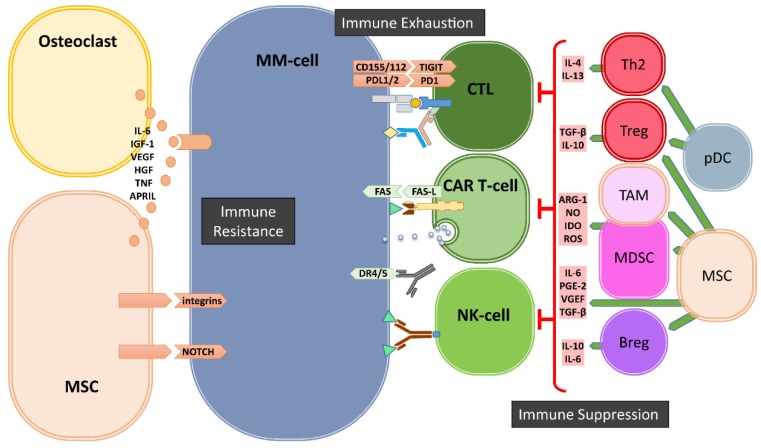
Bone marrow (BM) microenvironment-mediated mechanisms of immune evasion. In the BM, the cells of microenvironment mediate the escape of MM cells from immune system via three major mechanisms: immune suppression, immune exhaustion and immune resistance. Regulatory T- and B-cells (Tregs and Bregs), myeloid derived suppressor cells (MDSCs), Tumor associate Macrophages (TAMs), dysfunctional dendritic cells (pDCs) as well as mesenchymal stromal cells (MSCs) and osteoclasts generate a highly immune suppressive environment to suppress T- and NK-cells. Immune exhaustion is the result of the upregulation of immune checkpoints such as PD1, TIGIT on immune cells and their ligands on MM cells. The third mechanism of immune escape is the development of resistance against cytotoxic killer mechanisms of immune effector cells mediated by soluble factors and especially by cell–cell contacts between MSCs and MM-cells.

**Figure 2 cancers-12-00988-f002:**
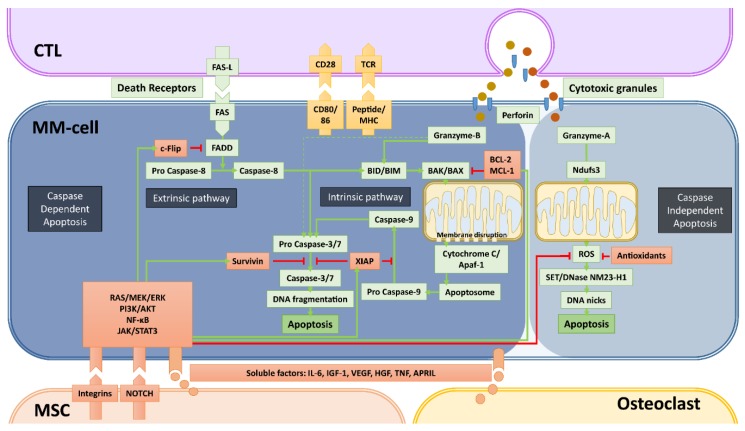
Caspase-dependent and -independent apoptosis pathways utilized by cytotoxic T-cells (CTLs) and NK-cells and mechanisms of bone marrow microenvironment-mediated immune resistance. Apoptosis, the programmed cell death that is a consequence of DNA damage in the nucleus, occurs through caspase-dependent and -independent mechanisms. Cytotoxic immune cells such as CTLs and NK-cells (not depicted) utilize both mechanisms to kill cancer cells. The most important elements of both caspase-dependent and -independent pathways are depicted with green backgrounds. The molecules depicted with red backgrounds are the negative regulators of these pathways. Green arrows depict the activation, red arrows depict the inhibition/inactivation of the indicated molecules/processes. In caspase-dependent apoptosis, the DNA-fragmentation is executed by caspase-3 and -7, which are activated through two main pathways: i) the Extrinsic Pathway that is initiated by triggering of death-cell receptors and is mediated by caspase-8, and ii) the Intrinsic pathway that involves mitochondrial destabilization and is initiated by activation of BIM/BID by caspase-8 and granzyme-B. CTLs and NK-cells mediate the caspase-independent apoptosis mainly by granzyme-A, which causes the release of ROS through activation of the mitochondrial Complex I Protein Ndufs3. In this pathway, the enzyme NM23-H1 ultimately causes single stranded DNA damage by making DNA nicks. The indicated soluble factors produced by MSCs and osteoclasts, but mainly the stroma-MM cell interactions via integrins and NOTCH can significantly upregulate the negative regulators of apoptosis via the activation of the indicated survival/proliferation pathways. The final result of these interactions between MM- and accessory-cells is the development of resistance against the cytotoxic machinery of immune killer cells, in a similar fashion how drug resistance is induced.

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
