# Peer review of "Challenges for Immunotherapy in Multiple Myeloma: Bone Marrow Microenvironment-Mediated Immune Suppression and Immune Resistance"

_cancers, 2020, doi:10.3390/cancers12040988_

Round 1

Reviewer 1 Report

This is very nice timely review concerning one of the major field of pathobiology of MM with many implications for disease treatment.

The manuscript is well written, clear to the reader. Figures are of high quality.

I do not have specific comments

Author Response

We thank the reviewer for carefully reading the manuscript. 

We have done our utmost to correct the language and spelling errors.

Reviewer 2 Report

This paper is an excellent general discussion of bone marrow Microenvironment-mediated immune suppression and immune resistance in multiple myeloma, which is very clear and to the point. Therefore, there are no amendments, but I would like the author to add the following points.

1. Add to the introduction the proteasome inhibitor and the Histone deacetylase inhibitor, which are therapeutic agents for MM.

2. Please be specific about the “dual targeting strategie” in "8. Conclusions and future directions"

Author Response

We thank the reviewer for critically reading the manuscript. Below are our responses to his/her specific comments:

Comment 1

Add to the introduction: the proteasome inhibitor and the Histone deacetylase inhibitor, which are therapeutic agents for MM.

Response 1

We thank the reviewer for this comment.  We now added at lines 30-33  the use of specific proteasome inhibitors and histone deacetylase inhibitors as approved therapies for Multiple Myeloma.  To substantiate this statement we have added one more reference(ref 4):

Moreau, P. How i treat myeloma with new agents. Blood 2017, 130, 1507-1513.

Comment 2

Please be specific about the “dual targeting strategies” in "8. Conclusions and future directions"

Response 2

Thank you for comment. To meet this comment, without being too elaborative but mainly explanatory, we revised the sentence at lines 382- 386 as follows:

"Another option could be to improve the overall avidity of T-cells toward MM cells by endowing them with dual- or multiple CARs, whereby more than one single MM-associated antigen can be targeted [102]. Improving the interaction between T cells and MM cells this way can also result in more efficient lysis levels, which is required to overcome the BM-ME mediated resistance."      

We hope that this explanation will be sufficient to specify the idea of how dual-CART cells can be beneficial to overcome the BM-ME mediated immune resistance. 

Reviewer 3 Report

Title: Challenges for immunotherapy in multiple myeloma: bone marrow microenvironment-mediated immune suppression and immune resistance

Authors: L Holthof and T Mutis

Review: Cancers-761145

Comments: I am satisfied with the overall quality of this review and references used.

Author Response

We thank the reviewer for carefully reading the manuscript. We did our best to correct the language and all encountered spelling errors. 

Reviewer 4 Report

This review article describing immunotherapy and bone marrow microenvironment in multiple myeloma is well written. 

The topic is very interesting and worth to publish.